# A Citizen Science Approach to Determine Physical Activity Patterns and Demographics of Greenway Users in Winston-Salem, North Carolina

**DOI:** 10.3390/ijerph16173150

**Published:** 2019-08-29

**Authors:** Joshua R. Dilley, Justin B. Moore, Phillip Summers, Amanda A. Price, Matthew Burczyk, Lynn Byrd, Patricia J. Sisson, Alain G. Bertoni

**Affiliations:** 1Department of Family and Community Medicine, Wake Forest School of Medicine, Winston-Salem, NC 27157, USA; 2Department of Implementation Science, Wake Forest School of Medicine, Winston-Salem, NC 27101, USA; 3Department of Epidemiology and Prevention, Wake Forest School of Medicine, Winston-Salem, NC 27101, USA; 4Clinical and Translational Science Institute, Program in Community Engagement, Wake Forest School of Medicine, Winston-Salem, NC 27101, USA; 5Department of Exercise Physiology, Winston-Salem State University, Winston-Salem, NC 27110, USA; 6City of Winston-Salem Department of Transportation, NC 27101, USA; 7Citizen Scientist, Winston-Salem, NC 27101, USA

**Keywords:** citizen science, bicycling, physical activity, racial disparity

## Abstract

Citizen science is a growing method of scientific discovery and community engagement. To date, there is a paucity of data using citizen scientists to monitor community level physical activity, such as bicycling or walking; these data are critical to inform community level intervention. Volunteers were recruited from the local community to make observations at five local greenways. The volunteers picked their location, time to collect data and duration of data collection. Volunteer observations included recording estimated age, race or ethnicity and activity level of each individual they encountered walking, running or bicycling on the greenway. A total of 102 volunteers were recruited to participate in the study, of which 60% completed one or more observations. Average observational time lasted 81 minutes and resulted in recording the demographics and physical activity of a mean of 48 people per session. The majority of adult bicyclists observed were biking at a moderate pace (86%) and were white (72%) males (62%). Similar results were observed for those walking. We demonstrate the feasibility of using citizen scientists to address the current scarcity of data describing community-level physical activity behavior patterns. Future work should focus on refining the citizen science approach for the collection of physical activity data to inform community-specific interventions in order to increase greenway use.

## 1. Introduction

Citizen science is an emerging method of community engagement designed to increase relevance and improve translation of scientific research into action. The term “citizen science” was coined in the 1990s by Alan Irwin and Ricky Bonney and defined as “projects in which volunteers partner with scientists to answer real-world questions” [1]. Citizen scientists are involved in data collection over a period of time to address a specific question [2]. Citizen scientist approaches can allow for enhanced community engagement while simultaneously answering scientific questions that might otherwise be logistically challenging. The benefit of citizen science is often bidirectional as researchers gain data and volunteers increase knowledge or gain satisfaction from participating in research [3]. Changes in governmental policy have been noted as a result of several community citizen scientist projects [4].

While it is well known that American adults engage in insufficient levels of physical activity in general [5], there is substantially less information known about demographic trends among community greenway users, such as bicyclists and pedestrians. The limited national data available suggests bicyclists are most likely to be white, male, and of upper socioeconomic status [6,7,8]. Unfortunately, there is a paucity of literature addressing race, ethnic or socioeconomic status-specific barriers as it would relate to designing community-based interventions to improve cycling rates.

Organizations such as the National Bicycle and Pedestrian Documentation Project have been using volunteers to count the numbers of pedestrians or bicyclists to study activity for years [9]. Similarly, citizen science is well-suited for physical activity research due to the public nature of physical activity and because many types of physical activity rely on the built environment. Several studies demonstrate the effectiveness of citizen scientists at recognizing barriers in the physical environment that limit healthy activities and advocating for changes in the built environment that directly impact their health [10,11,12]. Some citizen scientist projects require physical activity in order to collect environmental data, which serves the dual purpose of increasing activity and simultaneously collecting scientific data [3]. Despite citizen science being an excellent methodology to study physical activity for the aforementioned reasons, there is a scarcity of literature reporting on physical activity patterns measured using a citizen science approach. These data are critical to inform community-level intervention and policy towards increasing positive health behaviors, particularly with regards to understanding the patterns of use and demographic subgroups of users in a given location.

There are known disparities in physical activity patterns in the U.S. [13,14]. Recently, the Wake Forest School of Medicine’s Clinical and Translational Science Institute’s (CTSI) Stakeholder Advisory Committee (SAC), comprised of leaders from local government, area universities and community-based organizations, convened to discuss disparities that persist within the local community. It was hypothesized that a citizen science approach would offer a way to collect data on the community-level physical activity patterns that would be otherwise difficult to obtain. Therefore, the purpose of the present study was to engage citizen scientists to collect information regarding the demographics and behaviors of the greenway users (both cyclists and pedestrians) in a mid-sized city in central North Carolina and compare the results to national reports of greenway users to see if disparities exist.

## 2. Materials and Methods

Our citizen science project was based in Winston-Salem—a city of nearly 250,000 in Forsyth County, North Carolina [15]. In Forsyth County, the population demographics are 57% white, 26% African American, 13% Hispanic, 2% Asian and 2% other [16]. Within Winston-Salem, there are five local greenways frequently used by cyclists and pedestrians (Table 2). This citizen project sought to characterize and define the demographics of the people cycling and walking in these five locations.

### 2.1. Recruitment

Volunteers were recruited largely from members of local bike advocacy organizations and Winston-Salem State University through social media groups on sites such as Facebook as well as through physical fliers posted in local bike shops and at the local university. Additional participants were recruited by promotion through social networking groups within the bicycling community in Winston-Salem. Most volunteers got involved in the study through social media Facebook groups, which was our most successful recruitment tool. The volunteers from social media websites were easiest to recruit.

### 2.2. Volunteer Training and Observation Recordings

The volunteers attended a two-hour training session that discussed how to make observations, the principles of ethical conduct of human subject research, Institutional Review Board (IRB) approval, and how federal regulations permit research projects to conduct observations in public areas without informed consent. The volunteers (citizen scientists) were trained about the specific locations at which to make observations. Volunteers were required to make observations from a set location at each of the five individual greenways. Citizen scientists were allowed to determine what time and what location they made observations at. There was no set times or days to make observations.

On the individual observation forms, volunteers were asked to record their location, the date, the start time for observing, and the end time for observing. The participants were advised to monitor the age, race or ethnicity, and activity level of the people walking, running, or bicycling past them. Age was defined into broad categories: youth (age 2–20 years old), adult (age 21–59 years old) and senior (60 years old and above). Race or ethnicity was defined as white, black, Asian, Hispanic, other or unsure. The activity level of observed people was described as either moderate or vigorous. Moderate walking was defined as moving at a slow, casual pace while vigorous walking was defined as jogging or running. Moderate biking was defined as a slower pace, less than 14 mph while vigorous biking was defined as more than 14 mph.

We used a validated observation method entitled the System for Observing Play and Recreation in Communities for the volunteers to follow while monitoring physical activity. Previous studies have demonstrated high inter-rater reliability and validity using this method across different domains with paid volunteers [17]. For the present study, we used a slightly modified version of the System for Observing Play and Recreation in Communities with an adapted protocol for use with greenways (an example of the observation sheet is available in the Appendix A). The method was modified by reformatting it for the online data collection system, limiting activities to walking or cycling, and modifying the scanning protocol to account for the nature of the greenway.

After acquiring their data, volunteers would total their numbers on paper then submit their results to an online portal (REDCap) [18]. Emails were regularly sent to participants over the data collection period in attempt to keep them involved.

### 2.3. Follow Up Survey

After data collection ended, an electronic survey was sent to everyone who completed the two-hour training. The survey attempted to elicit reasons for participating in the study and submitting observations. 

### 2.4. Statistical Analysis

At the end of the study, all data were aggregated and analyzed using descriptive statistics to report total number and proportions for each characteristic observed of greenway users. No additional analyses were performed.

## 3. Results

### 3.1. Citizen Scientist Engagement

A total of 102 volunteers were recruited to participate in the study to make observations at the greenways. Sixty percent of the 102 people who were at the training session completed at least one observation. Out of 61 observers, 28 (46%) performed only one observation, 13 (21%) completed two observations, and 20 (33%) completed three or more observations (see Figure 1). Of those who completed three or more observations, the median number of observations was 3.5 (range 3–22). Average observational time lasted 81 minutes and resulted in the recording of the demographics and physical activity of a mean of 48 people per session. The breakdown of observations by greenway site is depicted in Table 1. Approximately 58% of all observations were made at two of the five total sites. The majority of observations occurred mid-day (43%) between 11 a.m. and 3 p.m.. Only 26% of observations occurred before 11 a.m. and 31% occurred after 3 p.m.. The observations occurred almost equally among the days of the week. Approximately 30% of the observations occurred on the weekend (Saturday/Sunday), while the remaining observations occurred during the week (70%). Thursday had the lowest overall recorded observations (7%), while Tuesday had the most at 18%. 

### 3.2. Satisfaction Sruvey Results and Citizen Scientists’ Demographics

Of the 102 individuals who completed training, only 20 completed the survey (20% response rate). The respondents identified equally as men and women (50% each). The average age of the citizen scientists who completed the survey was 52 with ages ranging from 15 to 79. Most scientists identified predominately as white (84%). The remaining scientists were African American (15%) or identified as other (1%). Nearly all survey respondents (90%) agreed with statements like “I like the outdoors” and “I care about the environment” and “I enjoy outdoor physical activity such as walking, biking and running”. Nine (45%) of the respondents reported they did not use the greenways often. Approximately 90% reported “I wanted to learn about citizen science”. Only 13% of respondents reported they “were required to learn about citizen science for course credit”. Six reported they already engage in scientific activities in the community. The majority of survey responders reported they “wanted to give back to the community” (88%) and that they “bike/walk on the greenways” themselves (78%).

### 3.3. Greenway Users

Table 2 summarizes the gender, race, age and physical activity level of the greenwayer users recorded by citizen scientists. Of all the bicycle riders recorded, 86% were observed biking at moderate intensity. Only 14% of riders were cycling at a vigorous pace. The majority of adult moderate intensity bicycle riders were men (62%) and white (72%). Thirty-eight percent of adult moderate intensity cyclists were female. Of the remaining adult moderate intensity bicycle riders, 18% were black, 7% Hispanic, 2% Asian and 1% other. The majority of riders observed engaging in vigorous intensity cycling were men (85%) and white (63%). Only 15% of adult vigorous intensity bicycle riders were women and 25% were black, 2% Asian, 10% Hispanic and 0% other.

Similar patterns were observed for adult moderate intensity and vigorous intensity on foot. The majority of adults observed walking were walking at a moderate pace (91%), while only 9% were walking at a vigorous pace. Only 21% of moderate intensity adult walkers were black, 1% Asian, 3% Hispanic, 3% other and 51% were women. The remaining 72% were white and men (49%). Of the adults engaging in vigorous intensity on foot, only 11% were black, 1% Hispanic, 1% Asian and 1% other and 42% were female. The remaining 86% were white and men (58%). Table 3 displays physical activity based on ethnicity/race including all age categories. The results demonstrate more individuals, regardless of race, engage in moderate than vigorous activity.

Table 4 compares the observed data on pedestrians collected by citizen scientists to the census data in Winston-Salem. The census data closely correlates with the observed users on the greenways except for Hispanics being underrepresented and whites being overrepresented on the greenways. The Hispanic population constitutes 13% of Winston-Salem, but only 4% of total pedestrians were observed to be Hispanic. White greenway users made up 72% of observed pedestrians but are only 57% of the total population of Winston-Salem.

## 4. Discussion

In this project, we describe a citizen science approach to collect data on physical activity patterns and demographic information about greenway users in a mid-size city community in North Carolina. We focus the discussion first on the implications of the greenway data, and then discuss the limitations and potential of the citizen science approach in the context of collecting physical activity data within the community.

### 4.1. Greenway Data

We observed that white males were the most common demographic using the greenways in Winston-Salem. Our data suggest disparity across demographics of the greenway users in Winston-Salem, corroborating reports from other published data [13,14]. There is no data specifically available to North Carolina regarding the demographics of bicyclists or active pedestrians. However, the limited national data describing greenway users shows that bicyclists are most likely to be white, male, and of upper socioeconomic status [6,7,8], which is similar to our results. Unfortunately, few interventions exist to directly address health disparities as they manifest in physical activity. Of the limited research addressing racial or socioeconomic status barriers, it has been proposed that enhancing bicycle infrastructure could improve cycling rates [19,20]. Additionally, finding ways to improve safety by addressing the African American and Latino communities’ concerns about crime and racial profiling will be crucial to increase cycling [21]. 

While it is important to acknowledge that all racial and ethnic groups display considerable heterogeneity with regard to cultural norms, the literature on park/greenway use provides some relevant insights for the present study. For example, the literature suggests Hispanics’ value spending time with family and place less emphasis on physical activity or exercise when visiting trails and parks [22,23]. As a result, trails or parks may be a place of socialization for Hispanics but not for physical activity [24,25]. Similarly, Asians in the United States are more likely to visit parks with their extended family and were the least likely to go solo [22,23]. European Americans are most likely to visit alone and African Americans were likely to attend with a small group of close friends [22,23,26]. These social values could partially explain our observed findings that European Americans were present in a higher proportion on the trail, while Hispanics were present in lower proportion compared to the census data (Table 4). Social norms within ethnic/racial groups are important to consider when addressing disparities in physical activity. More work is needed to identify factors that may facilitate increased use of greenways across a range of demographic characteristics. 

### 4.2. The Citizen Science Approach

We demonstrated the ability to engage members of the community to obtain information about physical activity. Approximately 60% of the original 102 volunteers from the training session completed one observation. The few available recruitment rates reported from other citizen scientist studies suggest that rates may vary from 15–75% [27]. This limited data suggests our retention rate is within the expected range. The utilization of “email blasts” in the study likely boosted our participation or retention rate as previous studies have demonstrated [27].

Furthermore, our results are consistent with previous research demonstrating that a disproportionately large volume of observations come from a small percentage of volunteers. As a result, using these motivated volunteers to recruit other volunteers may be effective [28] and create a more productive, like-minded community for future studies. Finding a way to harness these highly engaged citizen scientists’ interest and curiosity may help increase participant turnout, involvement and volume of recordings. Based on our survey, most of our volunteers identified as caring about the environment or physical activity. Although the response rate was limited, the findings suggest discussions with volunteers to reinforce the impact of their data collection on their community could increase participation, as well as harness interest and curiosity [29,30]. Having a volunteer recognition program, which was not included in this study, could have boosted participation and retention as well [30].

The citizen science approach has several noteworthy limitations. While the citizen science method is in theory an excellent way to collect high volumes of data that may be otherwise inaccessible to researchers, this approach to data collection has raised concerns regarding the validity of the data [31,32,33]. This approach is challenged by having volunteers as opposed to employed research technicians as the citizen scientists are less open to systematic assignment to days, times, or locations for their observations. As such, greenway counts are representative of the objective traffic only when observers chose to collect data and their choices were not systemically assigned in this study which would significantly improve validity. Our results are likely biased as the majority of observations occurred at only two locations (58%) and observations occurred mostly during the week (70%). Our results likely capture a different demographic than the true population [34].

Previous studies have demonstrated that four days per week and four times per day of monitoring is sufficient for accurate monitoring of physical activity [35]. Having volunteers set out at certain times of day during certain days of the week would allow for more accuracy in determining the true population that uses the greenways. A systemic, regimented approach is key to obtaining valid and accurate results. Continuous bicycle counters can be a useful adjunct to help identify the volume of cyclists and pedestrians as increased length of observation increases the accuracy of the measurements. Counters can significantly reduce the variability of the calculation of annual average daily bicyclists when compared to short, hourly counts [36]. However, counters do not allow for the collection of demographic data, which is where citizen scientists can be of instrumental use.

Future studies can improve upon our study design further by implementing rigorous training of volunteers to ensure the accuracy of results. All of the data collected by our volunteers rely on subjective categorization or interpretation (e.g., intensity, race). In this study, there was no way to confirm the accuracy or reliability of the data. As a result, interpretation of our data is limited. Prior studies have used an expert to identify the accuracy of volunteer data [37], photo submission, or replication by multiple participants [38], and previous citizen scientists’ projects report the importance of providing clear directions or explanations during the tutorial [39]. In our study, only one training session was held prior to data collection that outlined the categorization. The ability of volunteers to identify gender, race and age was not evaluated in the present study, but a review by Evenson and colleagues reported good inter-rater reliability of direct observation instruments (e.g., average agreement >80%) in prior studies [40]. However, due to the novel nature of these assessments for citizen scientists, future studies should have training sessions focused on identifying pictures and videos of varying users in different physical activities to ensure accuracy of age, gender and race/ethnicity assessment. Having multiple training sessions during the collection period may help improve the accuracy of results and allow volunteers to be better equipped to provide accurate data.

As citizen science emerges as a tool to collect data on the community that is otherwise inaccessible to public health researchers, attention must be paid to the bioethical considerations that are associated with the new information that is revealed. This includes ways in which the information could be used or exploited outside of the original intents and the ways it may impact the relationships of citizen scientists to the larger public or private institutions with whom they collaborate [41]. It is imperative that future studies are aware of these bioethical concerns and create protocols to address these concerns including fully informing the citizen scientists before participation, especially in projects using geolocated data [42].

Despite the limitations facing citizen science, studies have demonstrated the validity of citizen science [43,44]. Citizen science has been validated in studying numerous biological species and answering several diverse large-scale questions such as snail shell polymorphisms [45], hunters monitoring wolf numbers [46], monitoring air pollution concentrations [47], water quality [48] and light pollution [49]. Critiquing our protocol with the above suggestions could lead to citizen science helping to answer questions about community physical activity.

This study has several strengths. We demonstrated the feasibility of using citizen scientists to collect otherwise largely inaccessible data about patterns of physical activity. While the overarching theme of researchers and volunteers collaborating defines citizen science, there are heterogeneous approaches to the extent of the partnership, varying from public involvement in just data collection to volunteer involvement in all aspects of the scientific process. We offer a clear protocol and several important suggestions to future citizen scientist projects to address limitations of the data as discussed above.

Moreover, we contribute to an important topic—environmental determinants of beneficial health behaviors—for which there is limited data available. Community-specific data collection is essential in order to understand the current and local patterns of physical activity and identify subgroups for interventions to improve public health outcomes. Previous studies have demonstrated policy change as a result of citizen science projects [4] and demonstrated the ability to improve the physical environment in order to encourage physical activity [12].

## 5. Conclusions

We show that citizen science holds promise as an approach to study physical activity since it allows for the mass aggregation of data. The limited data from this study suggests greenway user bicyclists are most likely to be white men, consistent with other studies. Our study outlines clear limitations of the citizen science approach with actionable items to improve data validity for future citizen science studies focusing on physical activity. More research is critical to identify points for future community-level interventions to facilitate equitable access to and use of greenways for physical activity.

## Figures and Tables

**Figure 1 ijerph-16-03150-f001:**
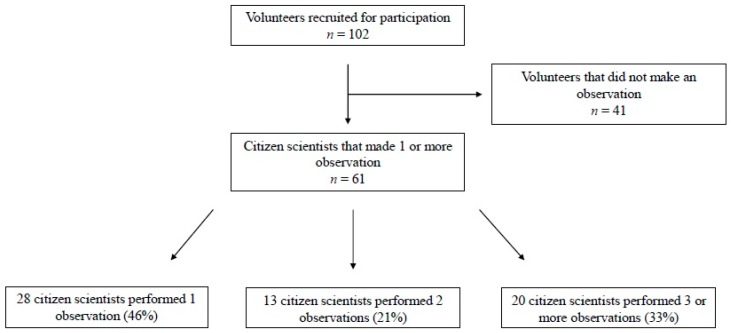
Volunteer recruitment and number of observations completed.

**Table 1 ijerph-16-03150-t001:** Total observations by greenway site.

Greenway Site	Percentage of Total Observations (%)
Salem Lake	32
4th Street Trail	26
Strollway Trail	18
Bushy Fork Greenway	12
West End Blvd Trail	12

**Table 2 ijerph-16-03150-t002:** Characteristics of greenway ssers in Winston-Salem, NC.

Observed Characteristics ^a^	Bikers (*n* = 1298)	Pedestrians (*n* = 6416)
	*n* (%)	*n* (%)
Gender		
Female	450 (35)	3239 (50)
Male	848 (65)	3177 (50)
Race/Ethnicity		
White	913 (70)	4616 (72)
Black	245 (19)	1321 (20)
Hispanic	69 (5)	235 (4)
Asian	54 (5)	103 (2)
Other	17 (1)	141 (2)
Age		
Youth (2–20 years)	189 (14)	700 (11)
Adults (21–59 years)	1009 (78)	5152 (80)
Senior (60 years or older)	100 (8)	564 (9)
Physical Activity Level		
Moderate	1115 (86)	5863 (91)
Vigorous	183 (14)	553 (9)

^a^ Observations were reported by 61 citizen scientists.

**Table 3 ijerph-16-03150-t003:** Physical activity level by race.

Race/Ethnicity, *n* (%)	Bikers (*n* = 1298)		Pedestrians (*n* = 6416)	
	Moderate Intensity ^a^	Vigorous Intensity ^a^	Moderate Intensity ^b^	Vigorous Intensity ^b^
All	1115 (86)	183 (14)	5863 (91)	553 (9)
White	799 (72)	114 (62)	4158 (71)	458 (83)
Black	197 (18)	48 (26)	1255 (21)	66 (12)
Asian	28 (2)	4 (2)	98 (2)	5 (1)
Hispanic	74 (7)	17 (10)	220 (4)	15 (3)
Other	17 (1)	0 (0)	132 (2)	9 (1)

^a^ Moderate walking was defined as moving at a slow, casual pace while vigorous walking was defined as jogging or running. ^b^ Moderate biking was defined as a slower pace, less than 14 mph while vigorous biking was defined as more than 14 mph.

**Table 4 ijerph-16-03150-t004:** Observed pedestrian demographics compared to census data of Winston-Salem.

Race/Ethnicity	Observed Pedestrian Demographics (%)	Forsyth County/Winston-Salem Demographics (%)
White	72	57
Black	20	26
Hispanic	4	13
Asian	2	2
Other	2	2

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
