# Peer review of "A Citizen Science Approach to Determine Physical Activity Patterns and Demographics of Greenway Users in Winston-Salem, North Carolina"

_ijerph, 2019, doi:10.3390/ijerph16173150_

Round 1

Reviewer 1 Report

Overall, in my opinion, this is a solid study worthy of publication. The two primary contributions are valuable data on demographics of greenway users,  an evaluation of the process of enlisting volunteer data collectors in an urban recreation context, and a summary of citizen science motivations for participating in recreation-related data collection. The manuscript is clear and well organized.

My primary suggestion is that I think the analysis related to the greenway data is underdeveloped. This is a very hard-earned data set that raises lots of interesting questions, but the results and discussion sections are relatively short and somewhat vague relative to this potential. As evidence, the entire discussion around this data set is only 168 words. It is apparent this manuscript is more waited towards lessons learned on citizen science, but this data set also has lots of value.

Here are some suggestions:

1) More fully compare greenway demographic data to city demographic data. Are groups over or under represented relative to urban demographic trends? For example, the biggest discrepancy from city stats to greenway stats is in the Hispanic population (14% of urban population but only 4-5% of greenway users). You Could calculate standard error/confidence intervals for greenway data to see how representative they are relative to the city demographics.

2) more fully engage the literature related to trends found in these data. You have one sentence about crime and racial profiling being a factor in underrepresentation of non-white population involvement in recreation, but there are many cultural and other factors and a broader literature on this. For example:

Cronan, M. K., Shinew, K. J., & Stodolska, M. (2008). Trail Use Among Latinos: Recognizing Diverse Uses Among A Specific Population. Journal of Park & Recreation Administration, 26(1).

Baas, J. M., Ewert, A., & Chavez, D. J. (1993). Influence of ethnicity on recreation and natural environment use patterns: Managing recreation sites for ethnic and racial diversity. Environmental Management, 17(4), 523.

3) Explore trends in physical activity level by age and race from this data. Even a second table that physical activity level by race would be intriguing.

4) You discuss the limitations of the sampling (58% of samples at two sample points and more weekday sampling) in the citizen science section, but could also be useful to explicitly discuss caveats related to the data set. For example, could there be potential demographic bias of users if samples were primarily in geographic areas of concentrated demographic groups?

In regards to the citizen science, my suggestions are more minor:

1) Which of your citizen scientist recruitment strategies were most effective?

2) Any lessons learned from the training sessions? How many? How often?

Some other minor suggestions:

* Paragraph starting on line 199 that begins “Table 1 depicts…” could use some wording clarification. It isn’t always clear what statistics apply to each demographic group.  

* Table 1: make it explicit that the number in parentheses are percentages.

Author Response

Response to Reviewer 1 Comments

Point 1: More fully compare greenway demographic data to city demographic data. Are groups over or under represented relative to urban demographic trends? For example, the biggest discrepancy from city stats to greenway stats is in the Hispanic population (14% of urban population but only 4-5% of greenway users). You Could calculate standard error/confidence intervals for greenway data to see how representative they are relative to the city demographics.

We appreciate the reviewer’s response and critique. We have added an additional table comparing the Winston-Salem census data to our observed data. Further discussion regarding observed differences is expounded on in the discussion.

Point 2: more fully engage the literature related to trends found in these data. You have one sentence about crime and racial profiling being a factor in underrepresentation of non-white population involvement in recreation, but there are many cultural and other factors and a broader literature on this.

We appreciate the reviewer’s response. We have reviewed and added several new papers including the ones suggested to our discussion.

The added discussion is included below:

While addressing safety will be of importance, other literature suggests Hispanics’ value spending time with family and place less emphasis on physical activity or exercise [22, 23]. As a result, trails or parks may be a place of socialization for Hispanics but not for physical activity [24, 25]. Similarly, Asians are more likely to visit parks with their extended family and were the least likely to go solo [22, 23]. European Americans are most likely to visit alone and African Americans were likely to attend with a small group of close friends [22, 23, 26]. These social values could partially explain our observed findings that European Americans were present in a higher proportion on the trail, while Hispanics were present in lower proportion compared to the census data (Table 4). Social norms within ethnic/racial groups are important to consider when addressing disparities in physical activity.

Point 3: Explore trends in physical activity level by age and race from this data. Even a second table that physical activity level by race would be intriguing.

We appreciate this response. We have added an additional table outlining physical activity level by race. This will hopefully help elucidate more of our data set.

Point 4: You discuss the limitations of the sampling (58% of samples at two sample points and more weekday sampling) in the citizen science section, but could also be useful to explicitly discuss caveats related to the data set. For example, could there be potential demographic bias of users if samples were primarily in geographic areas of concentrated demographic groups?

We appreciate this response. Unfortunately, we are not aware of the demographic data surrounding each of the individual sites and whether this biased the results.

Point 5: Which of your citizen scientist recruitment strategies were most effective?

We appreciate this response. We have added a sentence in the methods section to clarify this. “Most volunteers got involved in the study through social media Facebook groups, which was our most successful recruitment tool.”

Point 6: Any lessons learned from the training sessions? How many? How often?

We appreciate this response. We have expounded on this in our discussion. Unfortunately, there is no defined frequency or number of training sessions that have been demonstrated to improve data collection. Having multiple training sessions could be of potential benefit in addition to a rigorous initial training session.

The updated paragraph is included below.

In our study, only one training session was held prior to data collection that outlined the categorization. Volunteers’ abilities to identify gender, race and age was not evaluated. Future studies should attempt to have a training session(s) focused on identifying pictures and video of varying users in different physical activities to ensure accuracy of age, gender and race/ethnicity. Having multiple training sessions during the collection period may help improve accuracy of results. Rigorous training sessions will allow volunteers to be better equipped and to provide accurate data.

Point 7: Paragraph starting on line 199 that begins “Table 1 depicts…” could use some wording clarification. It isn’t always clear what statistics apply to each demographic group.  

Thank you for this response. We have reworked some of the framing in the paragraph to make it easier to follow.

 Point 8: Table 1: make it explicit that the number in parentheses are percentages

We appreciate this response. We have updated the table to make it easier to read.

Reviewer 2 Report

How could you possibly accurately describe the age, race or ethnicity and activity level of people walking, running or bicycling past the observers?  Race and ethnicity might be easier but if you are guessing at age - you could be so far off it would change the results dramatically.  Putting 14mph to judge pace of bicycling - just observing you can tell if it was 10 mph or 15 mph? Your statement that your results are likely biased (for many reasons) is absolutely true.  In the conclusions you say more research is critical to identify point for future community-level interventions......... Why not do it before you publish the paper you have here?

Author Response

Response to Reviewer 2 Comments:

Point 1: How could you possibly accurately describe the age, race or ethnicity and activity level of people walking, running or bicycling past the observers?  Race and ethnicity might be easier but if you are guessing at age - you could be so far off it would change the results dramatically.  Putting 14mph to judge pace of bicycling - just observing you can tell if it was 10 mph or 15 mph? Your statement that your results are likely biased (for many reasons) is absolutely true.  In the conclusions you say more research is critical to identify point for future community-level interventions......... Why not do it before you publish the paper you have here?

We appreciate the reviewer’s response and critique. We agree that the collection of data could be improved to ensure validity. In our discussion, we do mention this limitation. However, we have added additional text to further discuss the potential impact on the study results. 

The text added is: In this study, there was no way to confirm the accuracy or reliability of the data. As a result, interpretation of our data is limited.

The study team chose to submit the study for publication as it was perceived that the study offers multiple contributions to the present literature, with potentially the largest impact in the realm of citizen science methods for community health data. While the data collection methods may be optimized in the future to improve on scientific accuracy, this is a pilot study and thus offers a proof of feasibility for a new data collection methodology alongside specific aspects for improvement; these insights may help to improve future studies and offer a platform for a scientific conversation to continue to develop this methodology towards public health research.

Reviewer 3 Report

The article might benefit from citing one additional reference and adding a paragraph of additional discussion to address the bioethical issues raised by citizen science:

Wiggins A.  The rise of citizen science in health and biomedical research. Am J Bioethics 2019.

Author Response

Response to Reviewer 3 Comments:

Point 1: The article might benefit from citing one additional reference and adding a paragraph of additional discussion to address the bioethical issues raised by citizen science:

Wiggins A.  The rise of citizen science in health and biomedical research. Am J Bioethics 2019

We appreciate this response. We have included an additional paragraph discussing the bioethical issues raised by citizen science. We cite the paper suggested along with a paper by Bowser/Wiggins. The added paragraph is highlighted below:

As citizen science emerges as a tool to collect data on the community that is otherwise inaccessible to public health researchers, attention must be paid to the bioethical considerations that are associated with the new information that is revealed. This includes ways in which the information could be used or exploited outside of the original intents and the ways it may impact the relationships of citizen scientists to the larger public or private institutions with whom they collaborate [40]. It is imperative that future studies are aware of these bioethical concerns and create protocols to address these concerns including fully informing the citizen scientists before participation especially in projects using geolocated data [41].

Reviewer 4 Report

The primary aim of the study, stated in the introduction, is to test the feasibility of making physical activity demographics measures by volunteers citizens. The secondary objective seems to be an evaluation of the citizen science results (demographic characteristics of the greenway users) in comparison with validated national results of physical activity estimation. 

The introduction clearly presents the state of the literature on the topic of citizen science and why it is of interest in the field of physical activity. 

The method section should include a statistical part where the statistical tests and programs used to describe their results are presented. While the authors wrote in the introduction that they aimed at comparing their results to national reports of greenway users to see if disparities exist. However, the methods section does not present any methodological aspect related to this objective.

The discussion is well written and describes appropriately the results into context without overinterpreting. 

The main merit of this paper is to promote citizen science in the field of "physical activity estimation". This is promising but further studies are needed in order to actually validate this method in comparison with validated tools.   

Author Response

Response to Reviewer 4 Comments

Point 1: The method section should include a statistical part where the statistical tests and programs used to describe their results are presented. While the authors wrote in the introduction that they aimed at comparing their results to national reports of greenway users to see if disparities exist. However, the methods section does not present any methodological aspect related to this objective.

We appreciate the reviewer’s encouraging response and critiques. The paper was revised to include a separate section header for statistical methods. No additional statistical methods were performed as the primary focus of this paper was to offer a proof a feasibility for citizen scientist physical activity data collection along with specific aspects for improvement of future studies. We did add an additional table comparing the Winston-Salem census data to our collected data. The comparison was further highlighted in the discussion. We broadly compare our results mentioning the most common and least common demographics observed.

Round 2

Reviewer 2 Report

There is  not that much difference between this new edition and the original.

In this edition there is much more about how the study did not add to the information but rather explained why it was "likely biased, subjective" and the "volunteers' abilities to identify gender, race, and age was not evaluated."  The paper tended to stereotype African Americans and Latinos - Hispanics came to parts for socialization but not for physical activity.  One whole paragraph explained these stereotypes. Again, accurately determning the age of participants just by observing them go by would not be an easy task - good guess maybe.

Author Response

Response to Reviewer 2 Comments

Point 1: The paper tended to stereotype African Americans and Latinos - Hispanics came to parts for socialization but not for physical activity.  One whole paragraph explained these stereotypes.

We appreciate the reviewer’s concerns that we are stereotyping individuals by race or ethnicity. This was not our intent. We simply sought to introduce a body of literature that indicates that cultural differences between racial and ethnic groups exist with regards to park use. We have revised the language we use in the manuscript to better introduce the pertinent literature.

Point 2: Again, accurately determining the age of participants just by observing them go by would not be an easy task - good guess maybe.

While we did not assess reliability of the demographic assessments in the current study, there exist a number of studies that confirm that demographic information (e.g., age, race, gender) can be reliably assessed by observers. For example, in a review by Evenson and colleagues [Evenson KR, Jones SA, Holliday KM, Cohen DA, McKenzie TL. Park characteristics, use, and physical activity: A review of studies using SOPARC (System for Observing Play and Recreation in Communities). Prev Med. 2016;86:153–166. doi:10.1016/j.ypmed.2016.02.029)], “average percent agreement and correlation coefficients exceeded 80% and 0.80, respectively, for total number of people observed, age, gender, race/ethnicity, and physical activity.” We have added a sentence in the manuscript to introduce this evidence.